# Recent Advancements in Development and Therapeutic Applications of Genome-Targeting Triplex-Forming Oligonucleotides and Peptide Nucleic Acids

**DOI:** 10.3390/pharmaceutics15102515

**Published:** 2023-10-23

**Authors:** Yu Mikame, Asako Yamayoshi

**Affiliations:** Graduate School of Biomedical Sciences, Nagasaki University, 1-14 Bunkyomachi, Nagasaki 852-8521, Japan

**Keywords:** oligonucleotide therapeutic, triplex-forming oligonucleotide, peptide nucleic acid, antigene, genome editing

## Abstract

Recent developments in artificial nucleic acid and drug delivery systems present possibilities for the symbiotic engineering of therapeutic oligonucleotides, such as antisense oligonucleotides (ASOs) and small interfering ribonucleic acids (siRNAs). Employing these technologies, triplex-forming oligonucleotides (TFOs) or peptide nucleic acids (PNAs) can be applied to the development of symbiotic genome-targeting tools as well as a new class of oligonucleotide drugs, which offer conceptual advantages over antisense as the antigene target generally comprises two gene copies per cell rather than multiple copies of mRNA that are being continually transcribed. Further, genome editing by TFOs or PNAs induces permanent changes in the pathological genes, thus facilitating the complete cure of diseases. Nuclease-based gene-editing tools, such as zinc fingers, CRISPR-Cas9, and TALENs, are being explored for therapeutic applications, although their potential off-target, cytotoxic, and/or immunogenic effects may hinder their in vivo applications. Therefore, this review is aimed at describing the ongoing progress in TFO and PNA technologies, which can be symbiotic genome-targeting tools that will cause a near-future paradigm shift in drug development.

## 1. Introduction

The recent advancement of artificial nucleotides, as well as their applications, have enabled the development of nucleic acid drugs, such as antisense oligonucleotides (ASOs) and small interfering ribonucleic acids (siRNAs). Combined with emerging drug delivery system (DDS) technologies, therapeutic oligonucleotides are recently attracting increasing attention as new drug modalities. We recently summarized the physicochemical properties of these artificial modifications (ASOs and siRNAs) [1] and DDS technologies based on material symbiosis [2]. Messenger RNAs (mRNAs) are the target genes of these ASOs and siRNAs, and decreasing the number of disease-associated mRNAs can result in symptom relief, improved function, and prolonged survival for patients. However, even if a target mRNA expression is transiently decreased by ASOs or siRNAs, mRNAs would still be transcribed continually from genomic DNAs. Therefore, long-term administration is necessary, as a radical cure is unlikely. Theoretically, the direct suppression or modification of the target genes in chromosomal DNAs is more efficient and promising, and such a strategy can bring a paradigm shift in drug development. Sequence-recognition ability is key to targeting specific DNA regions. Thus, several genome-targeting tools have been developed to modify genome sequences or inhibit gene expression. For example, the growing literature on genome editing using clustered regularly interspaced short palindromic repeats (CRISPR)-associated protein 9 (CRISPR-Cas9) [3] has promoted the feasibility of applying gene-editing technology to therapeutics [4,5]. However, these technologies require exogenous enzymes, and this could cause several drawbacks, such as cost, handling, and the induction of immune responses [6]. Thus, the development of more symbiotic and facile methods is desirable. In this context, triplex-forming oligonucleotides (TFOs) and peptide nucleic acids (PNAs) exhibit significant potential as genome-targeting tools [7,8], considering that those aforementioned ASO and DDS modification methods [1,2] are also applicable to TFOs and PNAs. Both TFOs and PNA can bind to a DNA duplex in a sequence-specific manner (Figure 1), which enables the modulation of the target DNA functions (the structural features of TFOs and PNA are described in the following sections). This direct genome-targeting using such nucleic acids is often described as “antigene” in contrast to the word “antisense”. This method was originally demonstrated separately by Darvan and Hélène’s group by showing targeted dsDNA cleavage using an EDTA·Fe-conjugated TFO [9] and azidoproflavine-conjugated TFO, respectively [10]. The other early works (~2008) of the antigene method are well summarized in the excellent review article [7]. While these researchers have demonstrated the great potential of the antigene strategy, there are no FDA-approved TFO or PNA drugs to date, and many factors remain elusive, such as limitations in the sequence recognition ability of TFOs and PNA. As there is much progress in both the development and application of TFOs and PNA, this review article is aimed at summarizing the TFO and PNA technologies, focusing on the crucial findings and recent advancements in their development, which will make them more practical, as well as their therapeutic application and future perspective.

## 2. Triplex-Forming Oligonucleotides

### 2.1. Outline of the Triplex-Forming Oligonucleotide Technology

TFOs exhibit a short oligonucleotide sequence, which binds to their target DNA duplex by forming a triple-helix structure on the major groove of the DNA via the Hoogsteen (parallel triplex; Figure 1, up) or reverse Hoogsteen (antiparallel triplex; Figure 1, down) hydrogen bonds. In the parallel triplex, thymine (T) recognizes the adenine (A):T base pairs to form a T·A:T triad, and protonated cytosine, (C^+^), recognizes the guanine (G)·C base pairs to form a C^+^·G:C triad. On the other hand, A or T recognizes the A:T base pairs to form an A·A:T or T·A:T triad, and G recognizes the G:C base pairs to form a G·G:C triad in the antiparallel triplex.

Existing studies on the human genome have revealed that most annotated protein-coding genes in the human genome contain a minimum of one unique targeting site for TFOs (triplex target sites, TTSs) in the promoter or transcribed regions [11,12]; they also contain several TTS-mapping tools to facilitate the selection of target sequences for TFOs [13,14,15].

Various TFO-treatment mechanisms have been reported. First, TFOs were developed for antigene strategies targeting clinically essential genes, particularly oncogenes, such as c-myc, bcl-2, HER2, and Ets-2 [16]. TFOs can downregulate gene expression levels by inhibiting either the binding of transcription factors to promoter regions [17], the formation of the initiation complex (Figure 2a) [18], or the transcriptional elongation (Figure 2a) [19]. TFOs are also known to upregulate gene expression by either triplex formation at the repressor site [20] or introducing transcriptional activation domains into TFOs [21,22]. Recently, the Simmel group reported that the triplex formation at the promotor region can induce either inhibition or activation effect depending on the triplex motif (pyridine or purine), the position of the polypurine sequence (sense or template strand), and the location of the triplex with respect to the conserved sequences of the promoter suggesting that the inhibitory or enhancing can be controlled by rational design of a TFO [23]. Second, TFOs can be employed to induce DNA double-strand breaks (DSBs) [9], which can be explored for genome editing (Figure 2b) [24] or apoptosis induction (Figure 2c) [25,26]. Tiwari et al. investigated the DNA DSB mechanism, confirming that triplex formation perturbed DNA replication fork progression, which resulted in a fork collapse and provoked DNA DSBs [27]. Third, by tethering functional molecules to a TFO, these functional molecules can exhibit sequence-specific reactivity to the target DNA (Figure 2d). Particularly, the introduction of a crosslinking agent into TFOs has been widely investigated, considering that the crosslinking between a TFO and the target DNA enables irreversible triplex formation that can enhance the antigene activity of TFOs [28]. Crosslinking-driven DNA damage can also induce several DNA-repair events that can be exploited for genome-editing technology [29].

### 2.2. Advancements in the Engineering of Triplex-Forming Oligonucleotides

#### 2.2.1. Triplex-Stabilization Technology

As mentioned in Section 2.1, stable triplex formation requires consecutive polypurine sequences in the target dsDNA, as Hoogsteen hydrogen bonds can only be formed between the TFO and purine bases of the target dsDNAs. The pyrimidine bases in the polypurine strand of the target DNA duplexes cause significant losses in the thermodynamic stability of the triplex. Moreover, the required protonation of cytosine in the parallel motif (Figure 1) limits the in vivo application of triplexes. There are ongoing efforts to solve these intrinsic triplex-stability issues that restrict the target sequences. One such approach for overcoming these issues involves developing artificial nucleobases that can form several hydrogen bonds with the pyrimidine-base-interrupting site of the target DNA duplex [30]. Although the natural nucleobases G can form a G·T:A triad in a parallel motif [31], and another natural nucleobase, T, can form a T·C:G triad in parallel and antiparallel motifs [32], only one hydrogen bond exists in these interactions; the bond is not sufficient for stable triplex formation (Figure 3). In this section, we considered several examples of the aforementioned artificial nucleobases, as well as other approaches utilizing DNA-intercalating molecules, and introduced base-stacking interactions, focusing on essential findings and recent advancements. The conventionally applied modifications for other oligonucleotide drugs, such as ASOs and siRNAs, were not particularly considered in this review.

##### Artificial Nucleobases Recognizing Pyrimidine Base Interruption in the Target Double-Stranded DNA

Recognition of the thymine–adenine base pair in a parallel motif

Miller’s group pioneered T:A base pair targeting using an artificial nucleobase in a parallel motif. They developed a *N*^4^-(3-carboxypropyl)deoxycytosine nucleobase (Figure 3, **1**) for the recognition of a T:A base pair in a parallel motif. They predicted that its constituent carboxylic acid could interact with the 6-amino group in A of the T:A base pair. The ultraviolet (UV)-melting experiments of a 15-mer TFO containing **1** selectively recognized a T:A base pair in dsDNA. The melting temperature (*T*_m_) of this TFO (Tris buffer, pH 7.0) was 7 °C higher than that of the triplex comprising the G·T:A triad [33]. However, the T:A selectivity of **1** was dependent on the buffer component, and a Tris cation was necessary to induce the T:A selectivity of **1**, which might not be practical for in vivo conditions. The Sun’s group developed an artificial nucleobase, S, containing two unfused aromatic rings and an acetamide moiety that was attached to 2′-deoxyribose (Figure 3, **2**). S was incorporated in the middle of an 18-mer TFO, and its thermal stability was screened against all four possible base pairs. Although the selectivity of S between S·T:A (*T*_m_ = 50 °C) and **S**·C:G (*T*_m_ = 46 °C) was slightly diminished as compared to that of G between G·T:A (*T*_m_ = 45 °C) and G·C:G (*T*_m_ = 35 °C), the *T*_m_ value of the triplex containing the **S**·T:A triad (*T*_m_ = 50 °C) was very close to those triplexes without any pyrimidine base interruption (the thermal stability of triplex with T·A:T triad was 51 °C, and the triplex with C^+^·G:C triad was 51 °C). They proposed that the aminothiazole part of **S** was key to T:A recognition, forming three hydrogen bonds between the T:A base pair in its proposed recognition scheme [34]. Recently, Ohkubo’s group reported a new artificial nucleobase exhibiting a quinoline skeleton, ^DA^Q_ac_ (Figure 3, **3**). The UV-melting experiments of the triplex consisting of the 18-mer TFO containing ^DA^Q_ac_ and a hairpin dsDNA revealed that this nucleobase recognized both T:A and C:G base pairs with *T_m_* values (pH 7.4) that were 13 °C higher than those of the A:T and G:C base pairs, respectively [35]. Although the number of artificial nucleobases for T:A recognition in a parallel motif is still limited, further structural refinement based on the new skeleton, such as ^DA^Q_ac_, is expected to improve both affinity and selectivity for a T:A base pair in the future.

Recognition of the guanine–cytosine base pair in a parallel motif

Brown’s group reported several artificial nucleobases exhibiting *N*-methylpyrrolo [2,3-*d*]pyrimidine-2(7*H*)-one as their core structures. They derivatized the 6-position of this core skeleton to introduce additional functionality. They noted that the nucleobase ^A^P (Figure 3, **4**) was the optimum analog for the C:G recognition in both affinity and selectivity [36]. Seidman’s group developed one of the most suitable nucleobases for C:G recognition to target the chromosomal Hprt gene sequence with a single C:G interruption. Combined with 2′-*O*-methyl modification, *N*^4^-(2-guanidinylethyl)-5-methylcytosine nucleobase (Figure 3, **5**) emerged as the best in terms of affinity and selectivity to the C:G base pair. The thermal stability of the triplex was 15 °C higher than that of the T·C:G triad containing a canonical triplex. This research demonstrated that the guanidine unit is a promising counterpart of G in the C:G base pair [29]. Further, Obika’s group developed an artificial nucleobase, GP^B^ (Figure 3, **6**) [37], containing a guanidine unit as a counterpart of G in combination with a locked nucleic acid (LNA) that is a nucleotide analog bearing a 2′-*O*,4′-*C*-methylene linkage. The partial introduction of LNA into an TFO is known to stabilize the triplex by inducing a preorganized helical structure for triplex formation [38]. The sequence selectivity of the GP^B^-containing 15-mer triplex was high, and its thermal stability was 15 °C higher than that observed using a T·C:G triad.

Recognition of the thymine–adenine base pair in an antiparallel motif

In 2020, Taniguchi and Sasaki’s group developed a C-nucleoside analog, AY-d(Y-R), bearing a pyrimidine skeleton (dY) and an amino-pyrimidine unit (AY) for the recognition of a T:A base pair (Figure 3, **7**). However, the association constants (*K*_a_) of the triplex formation depended on the neighboring bases, and the adjacent dG base at the 5′ side of the target, T, appeared to be required for stable triplex formation [39]. Recently, Ohkubo’s group reported an artificial nucleobase, 2-acetamido-6-aminoquinoline (^6DA^Q_ac_), for T:A-selective recognition (Figure 3, **8**). Generally, G-containing antiparallel TFOs were self-assembled, forming other higher-order structures, such as G-quartet, at high salt concentrations. Therefore, their binding ability to the target dsDNA was significantly reduced. However, ^6DA^Q_ac_ formed a triplex structure at low and high salt concentrations, representing valuable features for biological applications [40].

Recognition of the cytosine–guanine base pair in the antiparallel motif

Recently, Taniguchi’s group reported several artificial nucleobases for detecting C:G-interrupting sequences [41,42,43,44]. They designed 2-amino-4-methoxypyridinyl pseudo-dC (^4OMe^AP-ΨdC) with a high p*K*_a_ value (p*K*_a_ = 7.5) in the 1-*N* position (Figure 3, **9**). This nucleobase exhibited high affinity and selectivity for the C:G base pair, independent of the adjacent base pairs. The TFOs bearing this nucleobase formed a stable triplex with the hTERT and Cyclin D1 gene sequences containing multiple (3–4) C:G-inversion sites [41]. Aminoethyl-dAN is another artificial nucleobase possessing an amino-nebraline (AN) skeleton (Figure 3, **10**); it detected ^5m^C:G and C:G. Triplex stability depends on the adjacent bases, although the affinity to ^5m^C:G was higher than to C:G, demonstrating better ^5m^C:G recognition over C:G recognition [43]. A novel artificial nucleobase, 2-guanidinoethyl-2′-deoxynebularine (guanidino-dN), was developed based on **10** (Figure 3, **11**). The nucleobase (**11**) exhibited high binding affinities to ^5m^C:G and C:G base pairs in all four combinations of the adjacent bases. They demonstrated the utilities of the TFOs containing this nucleobase by inhibiting the activity of the ten–eleven translocation enzyme, which oxidizes the ^5m^C:G base pair in dsDNA [44].

##### Other Triplex-Stabilization Techniques Using Artificial Nucleobases

Cytosine mimic that can form hydrogen bonds with the guanine–cytosine base pair in a parallel motif

A natural parallel TFO requires low-pH conditions (pH < 6.0) owing to the low p*K*a value of the *N*3 position of C (p*K*a = ~4.5) (Figure 1, C^+^·G:C triad) [45]. Several C analogs were developed with higher basicity than C (5-methylcytosine, 5-mC) or with proper hydrogen-bonding groups (pseudoisocytosine; Figure 4, **12**) [45]. Among them, 6-amino-5-nitropyridine-2-one (Z) seemed to be the most practical for this purpose, as the thermodynamic stability of the triplex was unusually higher than that of the underlying duplex (Figure 4, **13**) [46]. Z is an uncharged C-glycoside mimic of C^+^ bearing a nitro group at the 5-position. The p*K*a value of the *N*3 position of Z is ~7.8, and this facilitates stable triplex formation in isolated and contiguous G:C base-pair sequences, even under basic conditions (pH 9.0), and enzymatic incorporation of Z was also demonstrated. It was assumed that the nitro group contributed to this stability by decreasing the epimerization of 1′-anomeric carbon, as well as the stacking interactions.

Stacking interactions

Sasaki’s group developed several W-shaped nucleoside analogs (WNA) exhibiting a bicyclic skeleton, as well as an additional aromatic ring. This aromatic ring was incorporated to increase the stability of the triplex via the stacking interaction. For example, WNA-βC (Figure 4, **14**) displayed the recognition ability of a C:G-interrupting base pair with some sequence limitation [47]. Ohkubo’s group also reported several artificial nucleobases containing a sulfur atom to increase base–stacking interactions. Further, 2′-deoxy-6-thioxanthosine (s^6^X) was developed for G:C base-pair recognition (Figure 4, **15**). The consecutive incorporation of s^6^X into TFOs targeting G:C-rich sequences resulted in a 50-fold stable-triplex formation compared to using unmodified TFO owing to stacking interactions [48].

DNA intercalation

In 1992, Dervan’s pioneering work demonstrated pyrimidine–purine recognition using an artificial DNA intercalator, D3 (Figure 4, **16**). D3 was originally designed to recognize the CG base pair via two hydrogen bonds. However, employing proton NMR studies, it was revealed that this recognition was the result of the DNA intercalation of D3 at the 3′-site of the target base pair [49]. A natural compound, such as daunomycin, was also explored for the stabilization of the triplex structure. Catapano’s group introduced daunomycin at the 5′-end of an GT-rich antiparallel TFO via a hexyl linker that was attached to the C-4 position of daunomycin (Figure 4, **17**). The daunomycin-conjugated TFO increased the stability of the triplex without compromising its specificity and reduced the transcription of the endogenous c-myc gene in cells [50]. Recently, Brown’s group reported the stabilization of triplex by thiazole orange (TO)-intercalator-introduced T-derivatives, TO_B6_-pdU (Figure 4, **18**). TFO containing three TO_B6_-pdU increased the thermodynamic stability of the triplex at pH 7 by 45 °C compared with an unmodified TFO. Moreover, employing 5-(1-propynyl)cytosine as a counterpart of the C:G-inversion site, a TO_B6_-pdU-modified TFO can form a stable triplex with these pyrimidine base interrupted sequences [51,52].

#### 2.2.2. Recent Advancements in Functional Triplex-Forming Oligonucleotides for DNA Targeting

By tethering functional molecules to a TFO, such molecules can exhibit sequence-specific reactivities toward the target DNA (Figure 2d). Particularly, the introduction of a crosslinking agent into a TFO has been widely investigated, as the crosslinking between the TFO and the target DNA enables irreversible triplex formation that can enhance the antigene activity of the TFO. Thus, crosslinking-driven DNA damage can induce several DNA-repair events that can be explored for genome-editing technology. Therefore, crosslinkable TFOs exhibit great potential as therapeutic oligonucleotides. In addition to crosslinking, certain metalloenzyme-tethered TFOs can directly cut DNA and are considered new genome-targeting tools. The recent advancements of these functionalized TFOs will be introduced in this section.

##### Targeted Crosslinking Using Psoralen-Conjugated Triplex-Forming Oligonucleotide

Psoralen is a representative crosslinking molecule. It is a furocoumarin natural compound that is produced by various plants. It has been used to treat skin pigmentation disorders [53]. Psoralen intercalates into the pyrimidine base junction of the DNA double helix. Under UV irradiation (365 nm), the furan-ring (the 4′ and 5′ positions) and pyrone-ring (the 3 and 4 positions) sides of psoralen can undergo a [2 + 2] photocyclization reaction between the 5 and 6 positions of the pyrimidine base to form a cyclobutene ring [54]. Generally, the first [2 + 2] photocyclization proceeds at the furan-ring side to obtain a diadduct product (Figure 5), as the pyrone-ring moiety of psoralen is necessary for photoactivation. The furan side of psoralen may form an electron donor–acceptor (EDA) complex with a pyrimidine base. This EDA complex facilitates the first [2 + 2] photocyclization of the furan ring; therefore, diadduct formation is relatively unique to DNA. Several researchers have introduced psoralens into TFOs to perturb or induce DNA transcription or mutation, respectively. The chemical structures of the representative Ps–TFOs are shown in Figure 5. For the first time, Hélène’s group introduced psoralen into the 5′-end of a TFO in which the linker was attached to the 5-positions of psoralen via a C6 linker (Figure 5, **19**). Subsequently, they demonstrated in vitro sequence-specific transcription inhibition by the formation of a covalent bond at a promoter sequence [55]. Conversely, Glazer’s group demonstrated the first targeted mutagenesis of the λ-phage genome in bacteria using trioxsalen-conjugated TFO, **20**, in which 4′-hydroxymethyl-4,5′,8-trimethylpsoralen was attached to the 5′-end of TFO via a two-carbon (C2) linker arm [56].

They also investigated the effect of the length of the linker of Ps–TFO on the mutation frequencies at the targeted site, demonstrating that the C6 linker outperformed the C4 one (14% and 3%, respectively) [58]. Seidman’s group demonstrated several examples of endogenous genome editing of mammalian cells, such as gene knock-in [59] and sequence conversion [60], via the homology-directed repair pathway by coinjection of the single-stranded DNA donor with **20** [59,60]. Our group applied **20** to the detection of 5-mC, which is an essential epigenetic marker related to cell differentiation and cancer development. By employing **20** in combination with an artificial nucleic acid chaperone (PAA-g-Dex: poly(allylamine)-graft-dextran), **20** was selectively crosslinked to dsDNA containing 5-mC [61]. Recently, our group conducted a close investigation on the triangular relationships between the structural differences of these Ps-TFOs, photo-crosslinking efficiencies, and biological activities. We demonstrated that the photo-crosslinking efficiencies of these Ps-TFO did not reach their plateau under cell irradiation conditions, highlighting the need to develop more reactive psoralen derivatives [62].

The psoralen unit of these Ps–TFOs was generally introduced via the phosphoramidite method, which requires an automated DNA synthesizer, strict anhydrous conditions, and is relatively difficult to operate. Succinimidyl-[4-(psoralen-8-yloxy)]-butyrate (Figure 5, SPB; ThermoFisher Scientific, Waltham, MA, USA) is commercially available. It facilitates the convenient introduction of a psoralen moiety into a TFO. However, we demonstrated that the linker position (8-position of psoralen) was not appropriate for crosslinking. Therefore, we developed two novel psoralen NHS esters (Figure 5, **21** and **22**) [57]. We demonstrated the facile preparation of these Ps–TFOs through the conjugation of a 5′-amino-linker-tethered TFO with **21** and **22**, observing an efficient crosslinking formation (the diadduct formations were 0%, 57%, and 63% for SPB, **21**, and **22** after UV irradiation for 30 s, respectively). This NHS ester can be used for other TFO analogs, such as PNA (discussed in Section 3), which recently exhibited considerable potential as a genome-targeting tool [63].

##### Targeted Crosslinking Using Platinum-Conjugated Triplex-Forming Oligonucleotides

Platinum compounds display DNA crosslinking abilities. A cancer chemotherapeutic drug, cisplatin, is a representative drug of this class (Figure 6). Platinum compounds primarily react with *N*-7 of purine bases, particularly G, to form a highly stable coordination complex, resulting in intrastrand and interstrand crosslinking in DNAs [64].

The structures of the reviewed platinum-compound-conjugated TFOs (Pt–TFOs) are summarized in Figure 6. In 2002, McLaughlin’s group reported TFO tethered with a *cis*-bifunctional platinated complex at the 5′ end of its T constituent (**23**) [65].

As the solid-phase-synthesis-based incorporation of the platinum complex in an oligonucleotide yielded an inactivated species, Pt–TFO was synthesized by a two-step procedure: (1) the phosphoramidite-method-based incorporation of a 2-(2-aminoethylamino)ethanol chelator and (2) metal complexation after the removal from the solid support. They revealed the sequence-specific delivery of the platinum complex to the target genes and the selective crosslinking using Pt–TFO. Conversely, transplatin-type conjugates (Figure 6) have been explored for this purpose. Miller’s group reported that the incorporation of *N*-7-platinated G-nucleosides (**24**) to the 3′- and 5′-ends of TFOs enabled TFOs to form highly stable adducts, resulting in interstrand crosslinking; the 2′-*O*-methyl analogs of these TFOs inhibited the transcription and replication of plasmid DNAs in cells [66]. Furthermore, they decreased the mRNA and protein levels of the endogenous human androgen receptor gene by 40% and 30%, respectively [67]. Recently, Kellett’s group reported a click-chemistry approach for functionalizing an alkyne-modified TFO (**25**) [68]. They developed several azido-bearing *cis*-platinum(II) complexes (**26**–**28**) and reported the modular synthesis of Pt–TFOs. Combined with TO, the aforementioned DNA-intercalating and triplex-stabilizing fluorophore (Figure 4, **18**) enhanced target binding and discrimination between target and off-target sequences.

##### Artificial Metallonuclease-Conjugated Triplex-Forming Oligonucleotides

Diverse metal complexes can cleave DNA. Combined with TFOs, such metal complexes acquire sequence selectivity and can become artificial nucleases and new genome-editing tools. As there is a comprehensive and excellent review on this topic [69], we only introduced recent advancements in this field here. In 2020, Hocek’s group developed several hybrids of Cu-chelated clamped phenanthroline (Clip-Phen) artificial metallonuclease (AMN) with TFOs (Figure 7, **29**) through the click-chemistry approach and achieved sequence-specific dsDNA cleavage. A Clip–Phen cupric complex can induce dsDNA cleavage in the presence of molecular oxygen and an external reductant, such as a thiol or ascorbate. These AMN-conjugated TFOs (AMN–TFOs), in which AMN was linked to the 5′-end or an internal T-base of the TFO through a flexible linker, facilitated a significant cleavage of the target duplex DNA (up to 34%) [70]. Kellett’s group also reported several AMN–TFOs [71,72,73]. They developed new copper-binding scaffolds, **5N_3_**-TPMA and 6N_3_-TPMA (TPMA = tris(2-picolyl)amine), that were designed as “caging” chelators to stabilize copper(II) ions (Figure 7).

These polypyridyl ligands were incorporated in the TFO through a click reaction, after which the purine-rich tracts of the green fluorescent protein gene were efficiently targeted [72]. They also demonstrated the enzymatic incorporation of DPA-modified uridine triphosphates (dU^DPA^TP) (Figure 7) into TFOs to develop AMN–TFOs that enabled the practical application of this AMM [73]. Combined with designer intercalators (Figure 7, DPQ or DPPZ) [71], the cleavage site was controlled toward achieving high-precision DNA cleavage [73].

### 2.3. Recent Advancements in Therapeutic Applications of Triplex-Forming Oligonucleotides

In 2011, Gillet’s group first demonstrated the effectiveness of methylphosphonate TFOs in targeting tumor necrosis factor-α (TNF-α). TNF-α is a proinflammatory cytokine that is crucial to the pathogenesis of many inflammatory diseases, such as arthritis. An anti-TNF-α TFO in a 0.9% NaCl aqueous solution was injected into acute- and chronic-arthritis model rats, and significant decreases were observed in the disease development in both models. Further, anti-TNF-α TFOs efficiently prevented synovitis, cartilage, and bone destruction in their joints. The TFO activity was significantly higher than that of the corresponding siRNA, indicating the criticality of direct gene targeting [74].

The amplification and (or) overexpression of the c-myc gene were associated with the poor prognosis or decreased survival of cancer patients. Recently, Huo’s group reported efficient and precise c-myc gene silencing using TFOs. They demonstrated the controlled release of a TFO using a gold nanoparticle-conjugated TFO (Au–TOF NPs) and the mediation of the self-assembly of ultrasmall AuNPs by another single oligonucleotide that exhibits a complementary sequence to the tail part of a TFO. Both oligonucleotides were assembled into large-size sunflower-like structures and disassembled under near-infrared (NIR) irradiation to release the “active” TFO. Tumor inhibition was studied using BALB/c nude mice. Tumor growth was synergistically controlled by the pre-incubation time and NIR-irradiation time point [75].

Recently, the therapeutic potential of TFOs against the HER2-gene-amplified breast cancer was reported [26,76]. Rogers’s group demonstrated the selective apoptosis induction of the HER2-amplified breast cancer cells by targeting amplified HER2-gene loci with TFOs. The four TTSs residing in intron 2, the promoter region, the coding region, and intron 19 were chosen as targets for TFOs, and the TFOs targeting intron 2 and intron 19 were the most effective. The DNA comet assay revealed that multiple triplex formations in amplified genes caused significant DNA damage through DNA DSBs, resulting in cell apoptosis responses, and it was found that the DNA damage was more severe in intron 2 and intron 19, consistent with their stronger apoptosis-inducing responses [26]. Notably, this apoptosis induction was only observed in HER2-gene-amplified breast cancer cells, and normal cells remained intact. Sequentially, the cancer-cell-selective anticancer activity was demonstrated using a mouse xenograft tumor model. The TFO encapsulated in the nanoparticles (NPs), composed of a copolymer of poly(lactic acid) and hyperbranched polyglycerol (PLA-HPG), was administered to mice, and a significant delay in tumor growth was successfully observed in a sequence-specific manner. No reduction in HER2 protein levels was observed in mice treated with the TFO, indicating that this anticancer mechanism is independent of HER2 activity. These results suggest that the anti-tumor effect of TFOs is derived from TFO-mediated DSBs. This research clearly demonstrated the effectiveness of targeting amplified gene loci using TFOs.

## 3. Peptide Nucleic Acid

### 3.1. Outline of the Peptide Nucleic Acid Technology

PNA was first reported by Nielsen’s group in 1991 [8,77]. It comprises an electrically neutral aminoethyl glycine backbone. PNA is an artificial DNA analog in which the negatively charged phosphodiester backbone is replaced by a charge-neutral pseudopeptide backbone (Figure 8). PNA exhibits several conformational flexibilities. It can adopt the A and B helical structures upon binding to target RNA and DNA, respectively, and form antiparallel and parallel duplexes. The antiparallel duplex is generally more stable than the parallel one. The charge neutrality of PNA enabled its binding to the complementary DNA sequence target with increased affinity and sequence specificity, resulting in the unique mode of PNA action as follows. Considering the high affinity of PNA to DNA, single-stranded PNA (ssPNA) can invade dsDNA (Figure 8). Pseudocomplementary PNA (pcPNA) [78] was designed to form two PNA/DNA duplexes. pcPNA contains artificial nucleobases, where 2,6-diaminopurine (D) and 2-thiouracil (sU) are used instead of A and T, respectively, to avoid PNA/PNA self-duplex formation by the steric hindrance between them. The bifunctional PNA [79] and tail-clamp PNAs (tcPNAs) [80] were designed to “clamp” one DNA strand comprising two sections that are connected by a flexible linker that enables the invasion of the target DNA duplex. This invasion is initiated by the triplex formation of one section through Hoogsteen base pairing, and the other section forms a Watson–Crick base pairing with the same DNA strand, resulting in a PNA/DNA/PNA triplex. This “clamp” distorts the DNA structure, followed by the recruitment of endogenous repair factors that can be explored for genome editing. Further, tcPNA contains an extended Watson–Crick binding section to create a “tail” for distorting the target DNA for an extended stretch with increased affinity to DNA. The aforementioned PNA designs have their advantages, and their effectiveness has been demonstrated in antigene and genome editing.

### 3.2. Advancements in Peptic Nucleic Acid Engineering

The unique characteristics of PNA, the aforementioned charge neutrality, and structural flexibility, also cause several drawbacks, such as low water solubility, poor cellular uptake, self-aggregation, and orientational ambiguity, in target-sequence recognition. Many modifications have been explored to overcome these drawbacks to improve the application potential of PNAs. As many comprehensive and excellent reviews have described these modifications and applications [77,81,82,83,84], we only introduced selected examples, focusing on essential findings and recent advancements.

#### 3.2.1. Modification of the Peptic Nucleic Acid Backbone

Ly’s group reported that the introduction of simple substituents, such as methyl, or hydroxymethyl, at the γ-position induced the preorganized, right-handed helical structure of the PNA backbone (Figure 9, **30**), resulting in strengthened binding to complementary DNA and RNA [85]. The same group demonstrated that the modification of the γ-position of PNA with guanidine, γ-GPNA (Figure 9, **31**), greatly improved the cellular uptake of γ-GPNA [86]. They also developed miniPEG γ-modified PNA (Figure 9, **32**) exhibiting superior nucleic acid binding due to the better preorganization of its PNA backbone. The miniPEG γ-modified PNA can invade any dsDNA sequence through only Watson–Crick base pairing to recognize the target [87]. Tahtinen’s group developed γ-guanidinylmethyl PNA (Figure 9, **33**) for recognizing triplex-forming PNAs. Three consecutive incorporations of **33** in PNA achieved the best binding affinity and Hoogsteen-face selectivity of the oligomer with improved cellular uptake [88]. Presently, the aforementioned γ-GPNA and miniPEG γ-modified PNA are widely used PNA derivatives for therapeutic purposes.

Appella’s group reported the detailed biophysical and structural properties of *S*,*S*-cyclopentyl PNA, cpPNA (Figure 9, **34**) [89,90]. The cyclopentane ring restricts the conformational flexibility of the PNA backbone, thus inducing a right-handed helix that favors binding to complementary DNA. Further, the affinity and selectivity improved with an increased amount of **34**, which enabled the customization of the stability of the complex. Recently, Ganesh’s group reported that the introduction of the gem-dimethyl (*gdm*) group influenced the *Z/E* rotamer ratio of the tertiary amide. The *α-gdm* monomer exclusively exhibits the Z-rotamer, whereas the *β-gdm* monomer exhibits the *E*-rotamer (Figure 9, **35**) [91,92]. Those *E/Z*-rotamers influenced the orientation preference of PNA in the formation of the complex. The same research group also reported aza-PNA bearing a nitrogen atom instead of a carbon atom at the α-position (Figure 9, **36**) [93]. Interestingly, the aza-PNA monomer assumed the *E*-form via an eight-membered hydrogen-bonded ring with backbone folding. A future study will discuss how this modification impacts its target recognition.

A novel class of PNAs, bimodal PNA, Cγ(*S/R*)-bm-PNA (Figure 9, **37**), was developed by Ganesh’s group [94,95,96,97]. Cγ(*S/R*)-bm-PNA contains an additional nucleobase at the γ-position of the PNA backbone, which enables bifacial recognition-forming duplexes at the B_1_ and B_2_ sides. The thermal stability of the DNA1/Cγ(*S/R*)-bm-PNA/DNA2 complexes was higher than those of their respective isolated duplexes [94,95]. The other bimodal PNA, bm-Cα-PNA (Figure 9, **38**), contains an additional nucleobase at the α-position of the PNA backbone [96]. Additionally, **38**, and **37** exhibited similar properties, although the target sequences were restricted to homothymine and homocytosine. These bimodal PNAs can be used to generate novel higher-order assemblies with DNAs and RNAs. For example, they reported a pentameric complex comprising a triplex and two duplexes, DNA1–Cγ(*S/R*)-bm-PNA/DNA2–Cγ(*S/R*)-bm-PNA/DNA3 [97]. Further investigations of these new bimodal PNAs, as well as their biological applications, are anticipated.

#### 3.2.2. Base Modification of Peptide Nucleic Acid

##### Base Modifications for Enhanced Triplex Formation

PNA forms a triplex structure via the Hoogsteen hydrogen-bond-forming T·A:T triad and C^+^·G:C triad base pairing. Therefore, the inability to form stable hydrogen bonds with the T:A and C:G base pairs, as well as the necessity of protonating C, limit their in vivo application, including parallel TFOs. Some of the designed base analogs for TFOs can be used for PNA. For example, the base analogs **2** [98] and **5** [99] (Figure 4) can be used for T:A and C:G base-pair recognition, and 5-mC, pseudoisocytosine (Figure 4, **12**) [100] can be used to replace C. Several artificial bases were originally developed for PNA [101,102]. Recently, Rozner’s group systematically surveyed simple nitrogen heterocycles for C:G base-pair recognition and observed that 3-pyridazinyl nucleobase, **P_N_** (Figure 10, **39**), forms more stable hydrogen bonds than the other heterocycles [102].

Thio-pseudoisocytosine (Figure 10, **40**) [103] was studied by Chen’s group as a replacement for C. Here, **40** forms stable base pairing through the synergistic effect of improved van der Waals contacts base stacking with hydrogen bond formation. Rozner’s group examined 2-aminopyridine M (Figure 10, **41**) as a more basic C nucleobase [104,105]. The replacement of six pseudoisocytosines in 9-mer PNA comprising six Ms increased the affinity of PNA to dsDNA by ≈100-fold owing to the cationic character of M.

##### Base Modifications for Peptide Nucleic Acid Functionalization

As mentioned in Figure 8, D and sU were designed to avoid the formation of an unproductive PNA–PNA duplex for pcPNA via a steric clash between the 2-amino group of D and thiocarbonyl group of sU (Figure 10, **42**) [106]. Hudson’s group reported an improved synthesis of sU, which eases the preparation of pcPNA and will accelerate future pcPNA studies [107]. Recently, Winssinger’s group reported a new pseudocomplementary G:C base pair for G-clamp-based dsDNA invasion (Figure 10, **43**) [108].

G-clamp is a phenoxazine-derived tricyclic C analog that can strongly bind to G through additional interactions, such as π-stacking, the electrostatic attraction of a positively charged amine, and hydrogen bonding at the Hoogsteen face [109]. The introduction of G-clamp into PNA significantly improved the thermodynamic stability of the PNA/DNA duplex [110]. They developed *N*-7 methylguanine (*N*7-Me-G), which was designed to cause steric and electrostatic repulsions between the G-clamp. The modified PNAs were used in the detection of the dsDNA target, the RT-RPA amplicon, from severe acute respiratory syndrome coronavirus 2 (SARS-CoV-2), with single nucleotide resolution, discriminating between two SARS-CoV-2 strains. Each strand of modified PNA exhibited fast strand invasion in the physiological condition and formed stable complexes with low equivalents of PNA.

An investigation of the Janus–Wedge triple PNA helix was pioneered by McLaughlin’s group [111]. Therein, bifacial artificial PNA nucleobases form hydrogen bonds with the Watson–Crick faces of the two DNA target strands. In 2018, Thadke’s group first reported a complete set of bifacial nucleobases that can distinguish the T:A, A:T, C:G, and G:C base pairs (Figure 10, **44**–**47**) [112,113]. The 6-mer miniPEG γ-modified PNA comprising these nucleobases efficiently and rapidly invaded target dsDNA with high sequence specificity under physiological conditions [113]. As **44**–**47** enable the targeting of any sequences, they could be novel gene-targeting tools, and their applications for therapeutic purposes will be investigated subsequently.

#### 3.2.3. Recent Advancements in Crosslinkable Peptide Nucleic Acid for DNA Targeting

Glazer and Nielsen’s group developed psoralen-conjugated pcPNAs, demonstrating that they can be used to induce single-base substitutions and deletions within the target site [114,115]. The linker was only linked to the 8-position of psoralen in these PNAs, making the effect of the linker position on crosslinking formation elusive. Recently, our group developed novel psoralen NHS esters (Figure 5, **21** and **22**), which enabled the easy conjugation of psoralen in the *N*-terminus of PNA and subsequent photo-crosslinking evaluation [57]. The yields of the adduct products of 5-Ps–PNA and trioxsalen–PNA prepared with **21** and **22** were 48% and 45%, respectively, after 30 s of irradiation; the yields were much higher than those of corresponding PNAs prepared using SPB (15%). The applications of these novel Ps–PNAs are being studied in our lab.

Crosslinkable furan-derived nucleobases were developed for PNA. The furan generates reactive species immediately after the activation of reactive oxygen species (ROS), which can be exploited for DNA crosslinking. Sequence-specific crosslinking was realized with γ-Lys-modified PNA [116,117]. Recently, Vilaivan’s group reported novel pyrrolidinyl PNA probes, exhibiting furan as a crosslinking formation with target DNA. The positional effect on crosslinking was examined, revealing that the external incorporation of the furan moiety improved the crosslinking. A future study will discuss its biological applications [118].

### 3.3. Recent Advancements in the Therapeutic Applications of Peptide Nucleic Acids

#### 3.3.1. Modulation of DNA Expression

Recently, Tonelli’s group reported the therapeutic applications of PNAs in targeting the MYCN gene [119,120]. The PNA employed in these studies was named BGA002, which is conjugated with a nuclear localization signal peptide [87] and specific to the MYCN gene. Although the sequence information was not disclosed, BGA002 exhibited more potent biological activities than their previous MYCN-targeting PNA (BGA001), which also exhibited antitumor activity in mice with rhabdomyosarcomas [121]. With PNA BGA002, they targeted MYCN-amplified neuroblastoma (MNA–NB) cells, in which MYCN was associated with increased ROS, downregulated mitophagy, and a poor prognosis. BGA002 inhibited the expression of the MYCN gene, causing profound mitochondrial damage through the downregulation of the mitochondrial molecular chaperone TRAP1; ROS increased with the concomitant decrease in the MNA–NB xenograft tumor. This research first described the relevance of MYCN in MNA–NB in mitochondrial maintenance [119]. In other research [120], the combinational use of BGA002 and retinoic acid (RA) was found to be beneficial in the treatment of MNA–NB. RA has been used to treat MNA–NB patients. However, RA resistance was observed for some patients. The coadministration of BGA002 and RA mediated the therapeutic efficacy of RA by inhibiting BGA002 on the MYCN gene. The inhibition of the MYCN gene by BGA002 decreased the mTOR pathway activity, followed by the autophagy response of MNA–NB. The efficacy of BGA002 treatment with RA was also demonstrated in an MNA–NB mouse model.

#### 3.3.2. Genome Editing

In 2016, Glazer et al. reported an in vivo correction of a β-thalassemia mutation in mice by miniPEG γ-modified tcPNA (γtcPNA) [122]. An γtcPNA and donor-DNA injection with poly(lacto-glycolic acid) (PLGA) NPs, as well as a stem-cell-factor treatment, ameliorated the disease phenotype and collected the β-globin gene by up to 7% in hematopoietic stem cells. In utero experiments have also been performed using the same mouse model, achieving improvements in the disease phenotype in pups [123]. It seems that the tcPNA works by opening the target dsDNA via strand invasion (Figure 8), which permits the donor DNA to hybridize the target DNA. Finally, the homology-directed repair using this donor DNA as a template resulted in gene collection [124]. In 2022, Piotrowski-Daspit’s group further demonstrated the utility of PLGA NPs encapsulating PNA miniPEG γ-modified tcPNA and donor DNAs in cystic fibrosis (CF) treatment. CF patients experience multiorgan dysfunction, which is caused by mutations in the CF transmembrane conductance regulator (CFTR) gene. The in vivo correction of the CF mouse model using γtcPNA resulted in a partial gain of CFTR function and improved the phenotype [125]. Recently, Glazer’s group also applied the same system to genome editing in single-cell embryos containing mutated eGFP genes. The blastocysts from the embryos that were treated with γtcPNA exhibited the expression of corrected eGFP with high editing levels, up to 94%. The mice from the re-implanted embryos consistently exhibited editing. This research is the first example of embryonic-gene editing using PNA [126]. In these above experiments, gene editing was very site-specific, with considerably low levels of off-target sequence modification.

## 4. Conclusions and Future Perspectives

Our group recently described the physicochemical properties of therapeutic oligonucleotide [1] and DDS technologies [2] from a symbiotic perspective. Currently, these factors are considered for RNA-targeting therapeutic oligonucleotides such as antisense oligonucleotides (ASOs) and small interfering RNAs (siRNAs). We anticipate that the same technologies can be considered in the development of genome-targeting antigene nucleic acids such as TFOs and PNAs, as introduced in this review, which offer conceptual advantages over antisense, as the antigene target is usually two gene copies per cell rather than multiple copies of mRNA that are continually undergoing transcription. Further, genome editing enables permanent changes in pathological genes, which will result in the complete cure of diseases. Nuclease-based gene-editing tools, such as zinc fingers, CRISPR-Cas9, and TALENs, are currently being explored for therapeutic applications, although their potential off-target, cytotoxic, and/or immunogenic effects may hinder their in vivo applications. Therefore, the potentials of TFOs and PNAs as symbiotic genome-targeting tools cannot be ignored. Although they exhibit great therapeutic potential, several shortcomings must be addressed to achieve the application of these nucleic acids, such as their limitations in the Hoogsteen-based recognition mode. As described in this review, recent advancements are overcoming these limitations, and several promising therapeutic applications have been achieved using TFO or PNA. The clinical use of these therapeutic antigene nucleic acids will cause a considerable paradigm shift in future drug development.

## Figures and Tables

**Figure 1 pharmaceutics-15-02515-f001:**
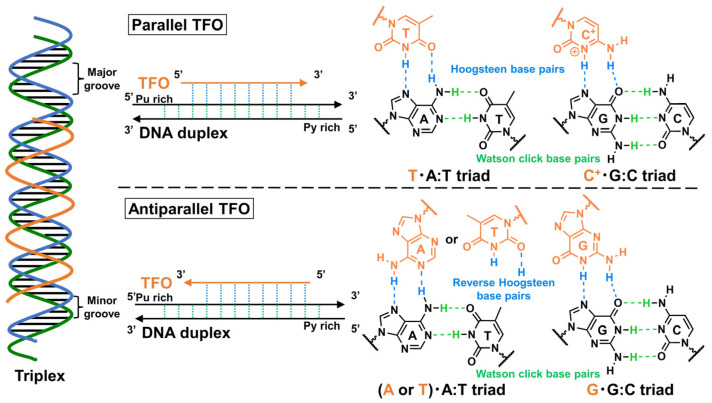
Structural features of TFO. The parallel and antiparallel triplexes (orange) are formed by Hoogsteen hydrogen-bond interactions (light blue) and reverse Hoogsteen hydrogen-bond interactions (light green), respectively.

**Figure 2 pharmaceutics-15-02515-f002:**
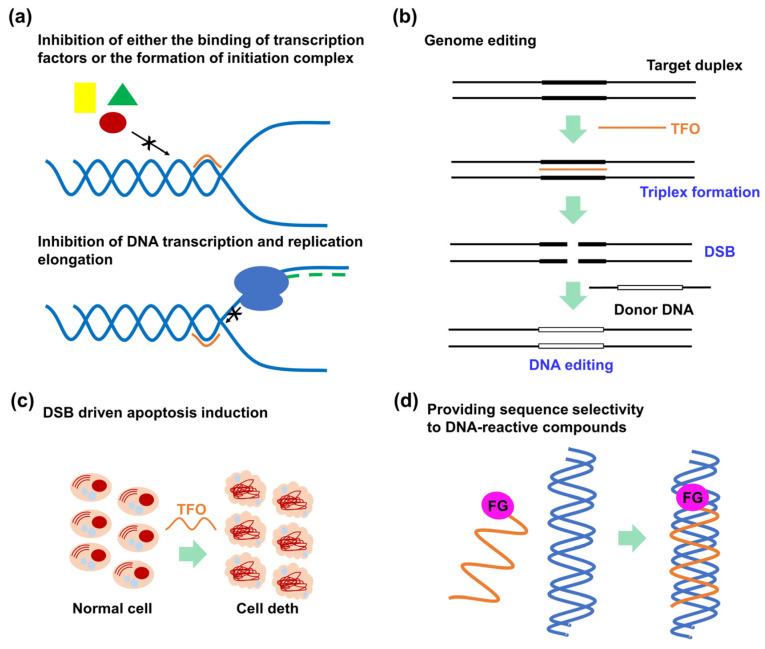
(**a**) Triplex formation perturbs DNA transcription and replication (the green, red, and yellow represent the transcriptional factors). (**b**) Triplex formation distorts the DNA duplex structure and induces DNA DSBs, which can be exploited for genome editing or (**c**) induce cell apoptosis. (**d**) TFO (orange) can provide a sequence selectivity to functional molecules (FG) that interact with ds DNA (blue).

**Figure 3 pharmaceutics-15-02515-f003:**
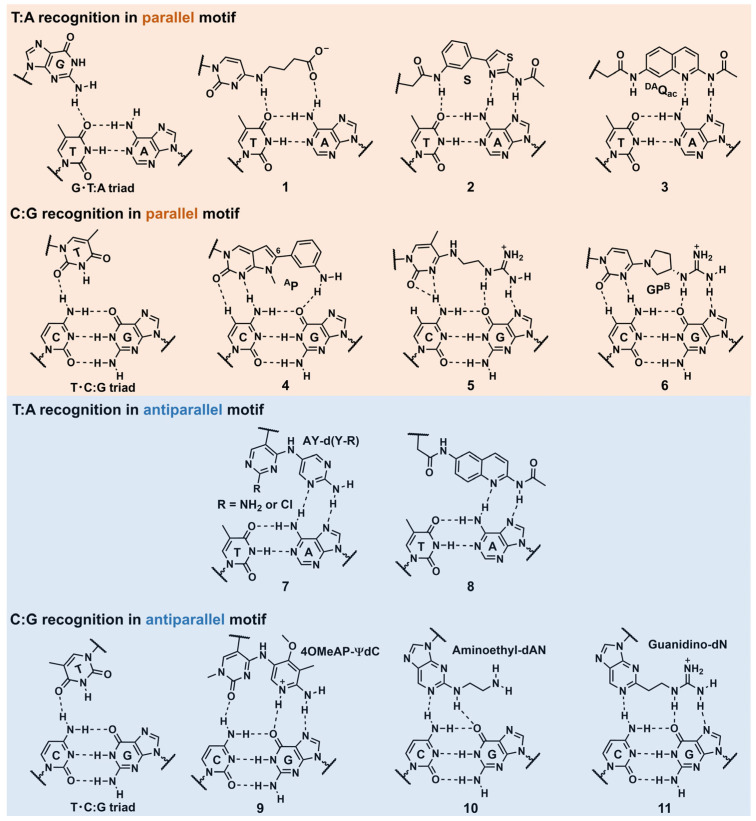
Selected examples of the artificial nucleobases developed for T:A and C:G inversions.

**Figure 4 pharmaceutics-15-02515-f004:**
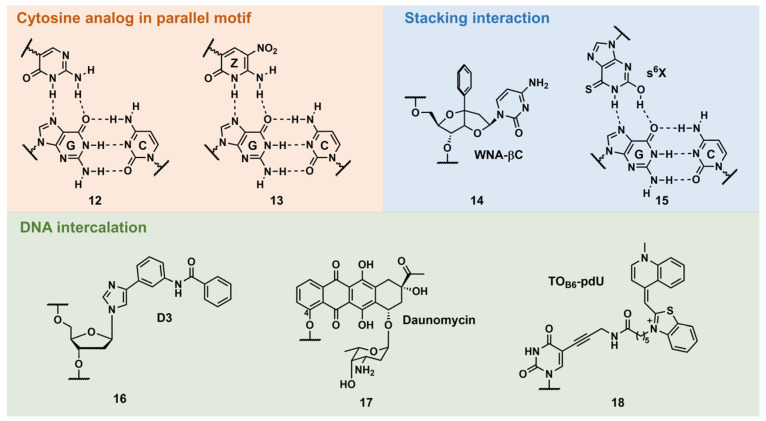
Selected examples of other types of triplex-stabilizing artificial nucleobases.

**Figure 5 pharmaceutics-15-02515-f005:**
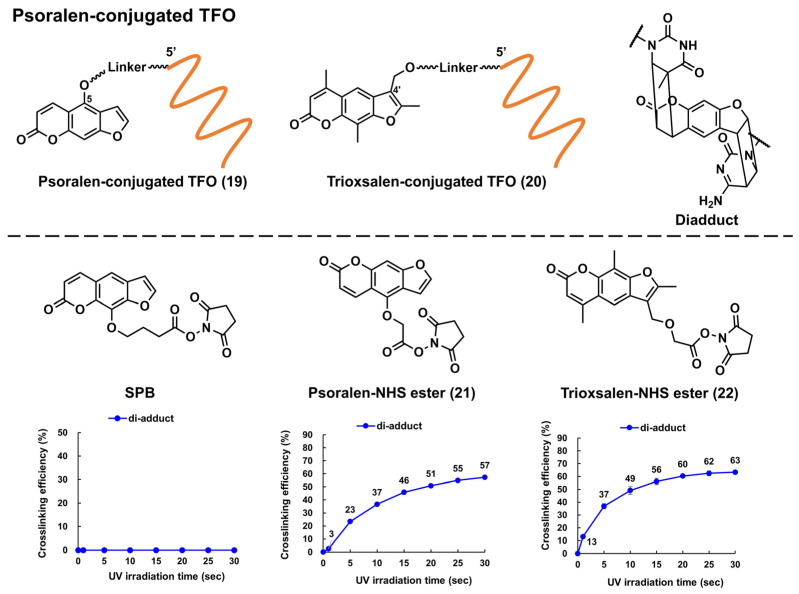
Structures of the psoralen-conjugated TFOs (Ps–TFOs) and psoralen *N*-hydroxysuccinimide (NHS) esters. The charts show the crosslinking efficiencies of the corresponding Ps–TFOs prepared using each NHS ester (the charts were quoted from [57]).

**Figure 6 pharmaceutics-15-02515-f006:**
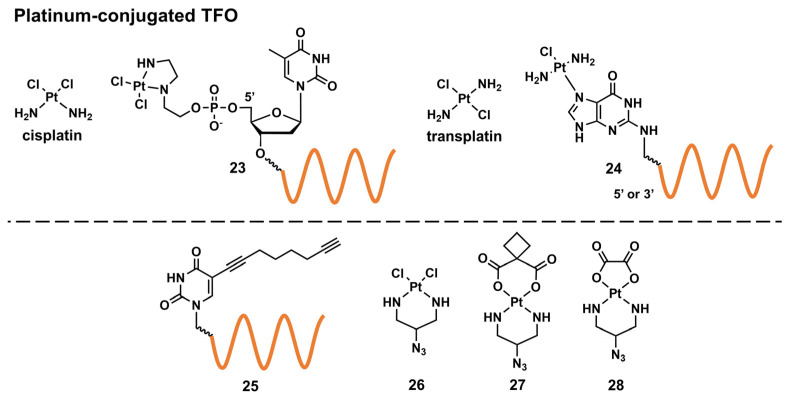
Structures of Pt–TFOs.

**Figure 7 pharmaceutics-15-02515-f007:**
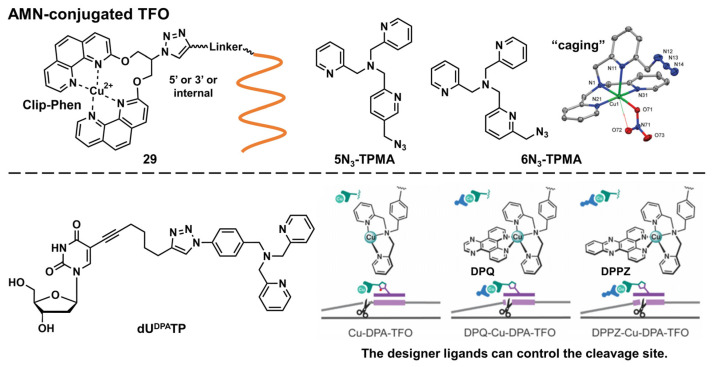
Structures of AMN–TFO (part of the figure was modified, following [72,73]).

**Figure 8 pharmaceutics-15-02515-f008:**
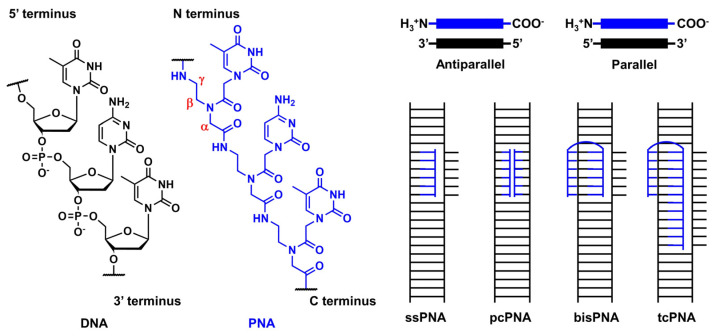
Structural features of PNA and unique mode of PNA actions.

**Figure 9 pharmaceutics-15-02515-f009:**
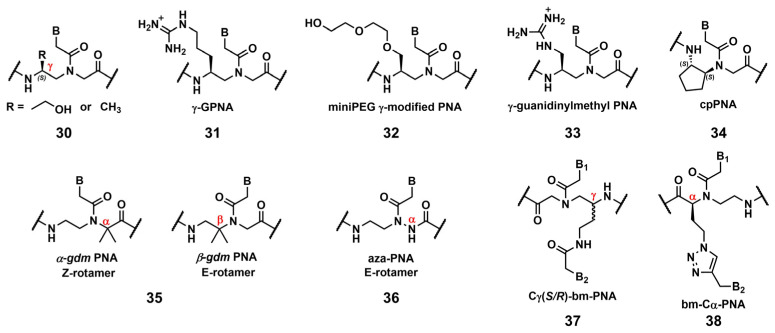
Chemical modifications of the PNA backbone. The position of the carbon is mentioned using α, *β*, γ (red).

**Figure 10 pharmaceutics-15-02515-f010:**
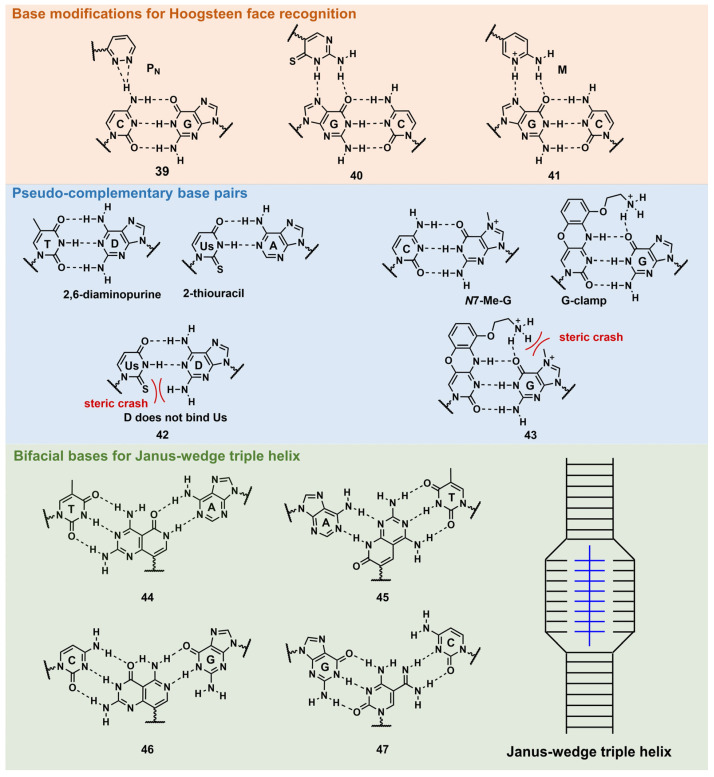
Nucleobase modifications of PNA.

## Data Availability

Not applicable.

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
