# Peer review of "Recent Advancements in Development and Therapeutic Applications of Genome-Targeting Triplex-Forming Oligonucleotides and Peptide Nucleic Acids"

_pharmaceutics, 2023, doi:10.3390/pharmaceutics15102515_

Round 1

Reviewer 1 Report

Yu Mikame and Asako Yamayoshi proposed a review article to Pharmaceutics, overlooking the topic of genom-targeting nucleic acids. The manuscript is well-written and structured, I was not able to decect inaccuracy and it involves relevant informations about the aimed field.

Comments:
About the intercalators and DNA damaging agents, it is known that they can cause mutagenesis. Is there any available data about, that conjugates of TFO-s with such molecules (e.g. platinum complexes) can causese undesired  cytotoxicity or other effects?

The anti-gene gene silencing as a potential use of the TFO-s is discussed in details, but I think it is also worth to mention, that triplex forming oligonucleotides (PNA for example) can also be used to induce the gene expression.

Summerized: I recommend the manuscript to be accepted after minor modifications.,

Reviewer 2 Report

The review by A. Yamayoshi and co-workers comes in accord with what may be seen as an upsurge of interest in DNA triple helices. The last overview of the topic, which this Referee, for instance, remembered was Duca et al, Nucleic Acids Res. 2008 Sep;36(16):5123-38 dedicated to the then 50 years’ anniversary of the discovery of the triple helix. Yet, the area of anti-gene therapy and triplexes was recently highlighted by several reviews (e.g. Li et al, Front. Pharmacol. 2022, 13, 1007723; Dalla Pozza et al, Chem. Sci., 2022, 13, 10193), to which the work of Yamayoshi et al comes as a very welcome addition. Now, as in 2022 65 years have elapsed since the publication of the seminal work of Alexander Rich and co-workers (Felsenfeld et al, J. Am. Chem. Soc., 1957, 79, 2023), another up-to-date review of the subject would be both timely and relevant.

I will be very glad to recommend a speedy publication of the paper pending a few reasonably minor revisions.

1. Firstly, the authors may wish to expand the Introduction part. The readers I believe would welcome some more information on the initial 25 years 1985-2001 of DNA triplex research of Claude Hélène and Peter Dervan with co-workers, and the origin of the concept of anti-gene therapy, to put more recent research into proper context.

2. In the description of antiparallel triplex (line 61 and further on), somewhat surprisingly, the triad TAT was not mentioned at all.

3. The authors describe extensive array of artificial nucleobases that accommodate pyrimidine interruptions X:Py:Pu in both types of DNA triplexes focusing on the increase in melting temperature but say very little on the binding specificity (e.g. X:Pu:Py). I think the specificity deserves some more comments as some of the mentioned modified bases were shown to be not too selective.

Very minor points

Line 9 – If ASOs is in plural, then siRNA should also be in plural – siRNAs, such as below in the lines 26-27. Also, please delete “These).”.

Line 31 – More precisely, “… (mRNAs) are the targets…” although non-coding RNAs such as micro-RNAs are becoming increasingly important targets as of recent.

Line 293 – Cisplatin is usually written without hyphenation. Same for transplatin.

Line 455, 460, 464 – Cytosine is the English spelling of the name; Cytosin (without ‘e’) is the German name. The same for its derivatives.

One small comment on the English of the paper. I think the authors in the introductory part used the word “symbiosis” and its derivatives somewhat too extensively. Additionally, the use of the word “aforementioned” as a verb (e.g. as in Line 468) is incorrect.

Reviewer 3 Report

Asako Yamayoshi et al. have submitted a detailed and comprehensive review on the most recent progress in triplex-forming oligonucleotides (TFOs) or peptide nucleic acids (PNA). An appropriate context is provided by including historical perspectives and reasonable comments used to distinguish between incremental progress and truly novel discoveries. This style raises the quality of the review above a mere collection of accomplishments to provide an educational and inspirational assessment of the field’s current state of the art. Contemporary problems in the field are described along with future goals and aspirations.

General points.

1.    The authors merely refer to literature results, without critical evaluation and selection, and therefore do not provide the reader with balanced view of the field and where it is going, in terms of the impact and significance of the numerous contrast agents reported.

2.    .      A thorough language editing by a native English-speaking person should be done.

Specific points

1.      The title of the manuscript is inappropriate because nucleic acid is quite broad concept and encompass lots of species but in the review only talked about triplex-forming oligonucleotides and peptide nucleic acid. To provide a succinct title would make the manuscript spread among potential readers.

2.      In the 2.2.1.1, the author discussed several exsamples with artificial nucleobases but majority of the word are to describe others’ work without further analysis and discussion. Some generality between the examples were welcome to be presented in each type of motifs.

3.      Is there any research using the PNA as skeleton to form the triplex?

4.      Therapeutic application part is too short. 

Minor editing of English language required
